# Willingness to work in rural areas and associated factors among graduating health students at the University of Gondar, northwest Ethiopia, 2021

Blen Getachew[1], Biruk Bizuneh[1], Birhanu Ewunetu[1], Dawit Kassahun[1], Dessalegn Fentahun[1], Destaw Ademe[1], Endeshaw Admasu Cherkos[2], Azmeraw Ambachew Kebede[3]*

1 Department of General Midwifery, College of Medicine and Health Sciences, University of Gondar, Gondar, Ethiopia, 2 Department of Women's and Family Health, School of Midwifery, College of Medicine and Health Sciences, University of Gondar, Gondar, Ethiopia, 3 Department of Clinical Midwifery, School of Midwifery, College of Medicine and Health Sciences, University of Gondar, Gondar, Ethiopia

* azmuzwagholic@gmail.com

**Data Availability Statement:** All relevant data are within the paper and its Supporting and Information files.

## Abstract

### Background

Many of the rural areas in developing countries are still in need access to quality healthcare services. To ensure the fair distribution of a high-quality health workforce and the availability of health services, there is a need to assess the background barriers that explain why healthcare providers are not interested to work in rural areas, thereby setting strategies to achieve universal health coverage. Therefore, this study is aimed to assess the willingness to work in rural areas and associated factors among health students at the University of Gondar.

### Methods

An institution-based cross-sectional study was conducted at the University of Gondar from August 15 to 25, 2021. A total of 422 study participants were selected using a stratified random sampling technique. A pretested self-administered questionnaire was employed to collect the data. Data were entered into EPI DATA 4.6 and exported to SPSS 25 for further analysis. Multivariable logistic regression analysis was performed to identify factors associated with students' willingness to work in rural areas. The level of significance was decided based on the 95% confidence interval at a p-value of $\leq 0.05$.

### Results

In this study, it was found that health students' willingness to work in rural areas was 78.4% (95% CI: 74.3, 82.4). Being male (AOR = 2.15; 95% CI: 1.17, 3.94), having intention to continue with their profession (AOR = 2.5; 95% CI: 1.28, 4.86), having a favorable attitude towards working in rural areas (AOR = 7.32; 95% CI: 5.71, 18.65), and having a mother with no formal education (AOR = 2.23; 95% CI: 1.02, 4.85) and completed primary education

**Funding:** The authors received no specific funds for this work.

**Competing interests:** The authors declare that they have no conflict of interest in this work.

**Abbreviations:** AOR, Adjusted Odds Ratio; CI, Confidence Interval; COR, Crude Odds Ratio; SPSS, Statistical Package for Social Science; SSA, Sub-Saharan Africa; WHO, World Health Organization.

(AOR = 2.69; 95% CI: 1.1, 6.61) were significantly associated with willingness to work in rural areas.

## Conclusion

The willingness of students to work in rural areas was optimal. This calls for concerned bodies to create a conducive environment for male and female students to engage in rural areas without hesitation. It is also important to ensure that students are willing to work in the rural areas voluntarily, instead of working in rural areas due to a lack of employment.

## Introduction

The health workforce plays a key role in the provision of health. The equitable distribution of quality health personnel contributes significantly to ensuring the availability of healthcare services, irrespective of location. Besides, it helps in progressing toward the Universal Health Coverage (UHC) goal by facilitating access to quality healthcare services for all [1].

More than 4 million health workers are needed to fill the gap in the global health workforce, as estimated by the World Health Organization (WHO) [2]. However, the geographical distribution of health workers is concentrated in affluent cities and regions worldwide, paying little attention to the rural areas. Regardless of the level of economic development and organization of the health system, the disparity in the distribution of health workers exists in almost every country in the world, but this problem is particularly prominent in developing countries [3].

The shortage of health workers in rural areas has been identified as one of the biggest challenges for most countries to achieve the desired health-related goals. Developing countries suffer severely due to a significant shortage of health workers proportional to the population [4]. Ethiopia is also one of the victims of this challenge, despite 84% of its citizen residing in rural areas [5]. One of the challenges to provide at least basic health services is the lack of trained health workers [6]. In connection with this, the WHO recommends that at least 4.45 doctors, nurses, and midwives per 1000 population are needed to achieve the minimum levels of key health interventions. However, Africa has a 2.2 health workforce per 1000 population, which is often challenging to achieve global goals [1].

In Ethiopia, the overall health professionals to population ratio were 1 per 1,000 population in 2018 [7], which is far below the minimum number estimated by the WHO. Hence, the healthcare system still suffers from the shortage mentioned earlier and the uneven distribution of healthcare workers between the urban and rural populations [8]. The shortage of health staff also has a significant impact on the healthcare delivery system such as the provision of essential life-saving interventions such as providing Expanded Program on Immunization (EPI), clean water, safe pregnancy, and childbirth services, and access to Human Immunodeficiency deficiency Virus/Acquired Immunodeficiency Syndrome (HIV/AIDS), tuberculosis (TB), and malaria treatment [9].

The Sustainable Development Goals (SDG) by United Nations (UN) member states in 2015, particularly goal 3 aim to ensure health and wellbeing for all. Target 3.1 specifically aims to reduce the Maternal Mortality Ratio (MMR) to less than 70 per 100,000, thereby saving the lives of one million mothers [10]. In addition, 405,000 people died of malaria in 2018, 94% of which occurred in Africa [11]. Moreover, an estimated 1.3 million TB-related deaths were recorded in 2017; additional 300,000 deaths among people living with HIV/AIDS [12]. All these burdens are more common in rural areas. Shortage and mal-distribution of healthcare

workers, particularly in rural areas will lead to failure to meet the SDGs. Hence, assigning adequate healthcare providers to these highly vulnerable groups will greatly help reduce the deaths of many lives.

Higher prevalence of mortality and morbidity are found in rural areas compared with urban areas [13]. Thus, the willingness of healthcare providers to work in rural areas is crucial to basically alleviating the problem. However, most health care providers are reluctant to work in rural settings. Some of the commonly cited reasons for not working in these areas are perceived low salary, unavailability of professional standards, unsatisfactory working environment, lack of access to skilled colleagues, and modern equipment [9]. Besides, some others stated that learning and training opportunities are scarce when they get employed in rural areas [14]. Furthermore, lack of the expected living conditions including electricity, water supply, and schools for their children, and having a plan to migrate out of their country after graduation are some of the reasons that lower the willingness to work in rural areas [14, 15]. Thus, it is important to give adequate remuneration to their work for health workers, better professional training, and educational opportunities, and identify effective and efficient human and non-human resources to meet the individual health needs of the underserved rural population.

Health science students' desire to go to the countryside will play an important role in the rural community that has been denied access to health services. Studies so far revealed that students willingness to work in rural areas were 52% in China [16], 45% in India [17], 33.5% in Nigeria [18], 55.4% in Ghana [19], and 29.7% in Ethiopia [20]. Being a male student [16, 19], having a family member of lower academic attainments and lower socioeconomic status [21], rural background [22], and having a good attitude towards the rural community [15] had a positive impact on students' willingness to work in the rural areas.

Although the health workforce in Ethiopia is growing, it is still less than half of the minimum number estimated by WHO. Increasing the willingness of graduating students by designing effective strategies will play a key role in reducing the burden of shortage of health care workers in the rural areas, thereby achieving the optimal health of the marginalized rural population. Besides, doing the research may offer basic evidence aimed at improving health workers' needs and setting strategies considering the rural community. Therefore, this study aimed to assess willingness to work in rural areas and associated factors among graduating medicine and health science students at the University of Gondar, northwest Ethiopia.

## Methods and materials

### Study design, period, and study area

An institution-based cross-sectional study was conducted from August 1 to 25, 2021. The study was conducted at the University of Gondar, College of Medicine and Health Sciences, which is found in Gondar city, northwestern Ethiopia. The college was established in 1954 and is considered one of the oldest and most famous Universities in Ethiopia. There are 12 fields of study in health and medicine such as Nursing, Midwifery, Clinical laboratory, Pharmacy, Anesthesia, Psychiatry, Health informatics, Physiotherapy, Optometry, Environmental and occupational health, Public health officer (HO), and Medicine. Currently, there are 1068 graduating students from the 12 departments by September 2021.

### Study population

All graduating health students during the study period were the study population.

## Sample size determination and sampling procedure

The sample size for this study was calculated based on the assumptions of the single population proportion formula by considering the following assumptions. Since there is no similar study, we used the proportion of student's willingness to work in the rural area-50% (p = 0.5), level of significance- 5% ($\alpha$ = 0.05), Z $\alpha$/2–1.96, and margin of error—5% (d = 0.05). Therefore, the sample size was calculated as follows

$$\text{n} = \frac{(Z\alpha/2)^2 * p(1-p)}{d^2} = \text{n} = \frac{(1.96)^{2*}0.5(1-0.5)}{(0.05)2} = 384.$$

After adding a 10% non-response rate, the minimum adequate sample size was found to be 422. There are 12 departments at the University of Gondar, college of medicine, and health sciences. All departments were included in the study. The lists of students were obtained from the registrar's office and the sampling frame was prepared by ordering the lists of students. Then, the total sample size was distributed proportionally to each department. Finally, the participants were selected using a stratified random sampling technique using a table of random generation (**Fig 1**).

## Variables of the study

The willingness of students to work in the rural areas was the outcome variable, whereas age, sex, place of origin, religion, professional category, paternal and maternal educational status, paternal and maternal occupation, and personal behaviors like exposure to addictive substances, attitude to work in the rural area, and plan to emigrate were the independent variables.

## Measurements

**Willingness to work in rural areas.** Students were asked whether they are willing to work in rural areas or not. A "yes" response was considered as willing to work in the rural areas [19].

**Favorable attitude towards working in rural areas.** Students' attitude toward working in rural areas was assessed using 11 questions: 1) Working in rural areas provide opportunities to use various skills 2) There is a supportive environment when working in a rural environment 3) Working in rural areas limits communication with professional peers 4) Working in rural areas provide opportunities to work independently 5) There is lack of amenities and entertainment in rural areas 6) People in rural areas are friendly 7) Working in these areas causes isolation from family and friends 8) Working as a health care provider in hospitals or health centers in rural areas is the most important contribution to the health of the population 9) Health science college prepared me well to work in rural areas 10) Working in rural hospitals areas is the most challenging 11) Working in rural hospitals provide opportunities for real-life problem-solving. Each question has five points Likert scale (1- strongly disagree, 2- disagree, 3- neutral, 4- agree, and 5- strongly agree). The minimum and maximum scores were 11 and 55, respectively. Thus, students who answered above the mean value are considered as having a favorable attitude [17].

## Data collection tools and procedures

The data collection tool was developed by reviewing the literature [16, 17, 19, 20] and data were collected using a structured, pretested, and self-administered questionnaire. The questionnaire was evaluated by experts before data collection. The questionnaire contains socio-demographic characteristics and academic-related characteristics. Six BSc in midwifery

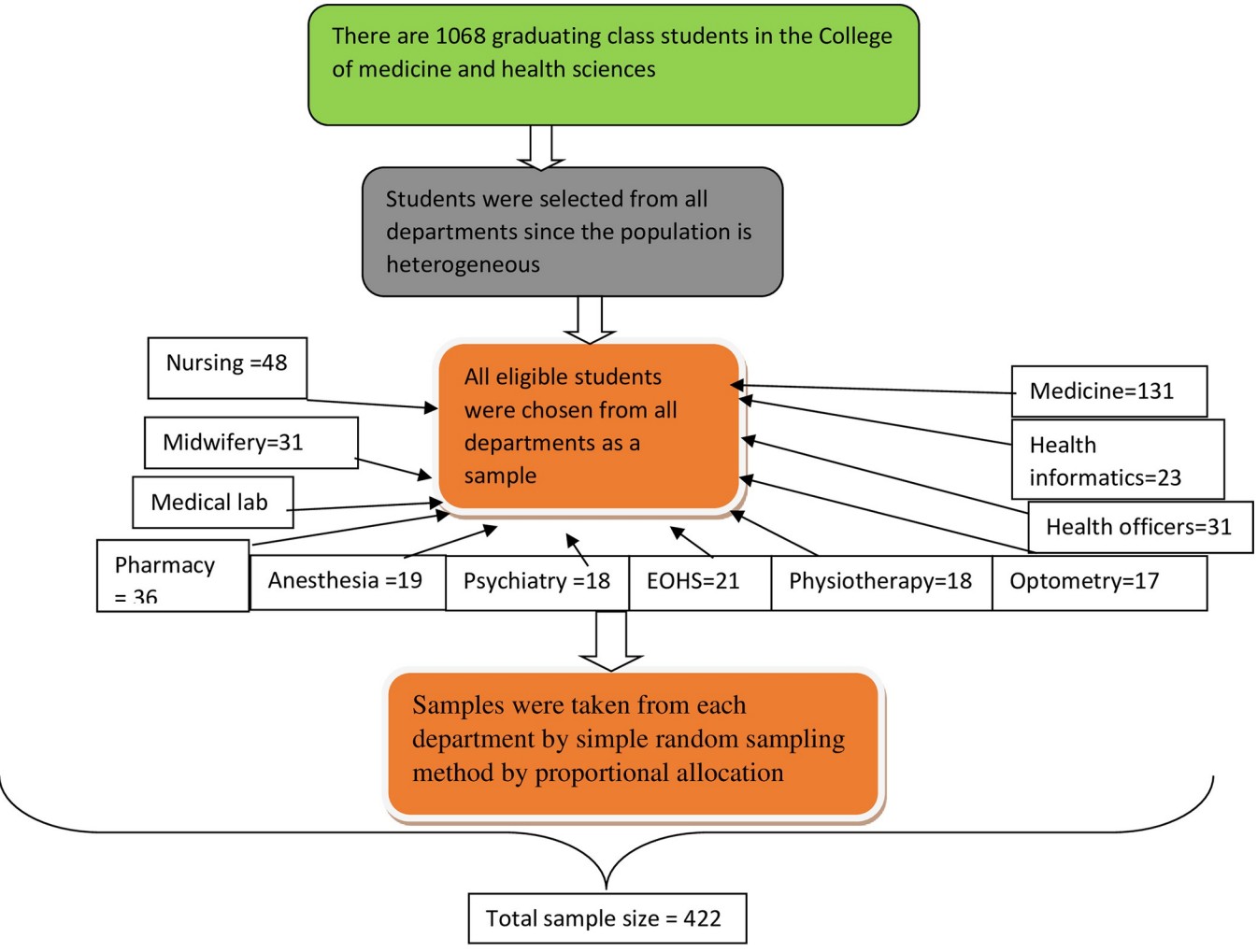

**Fig 1. Schematic presentation of the sampling procedure for the study done on willingness to work in rural areas and associated factors among graduating health students at the University of Gondar, 2021.**

graduating students and 1 MSc in Midwifery holder were recruited for data collection and supervision, respectively.

## Data quality controls

To ensure consistency and understandability, the questionnaire was initially prepared in English and translated to the local Amharic language and back to English. A pretest was done on 5% of the calculated sample size at Bahir Dar University to check the response, language intelligibility, and relevance of the questionnaire. The data collectors and supervisors were trained for a day regarding the overall data collection process. At the time of actual data collection, the collected data was checked daily by the supervisor for completeness.

## Data processing and analysis

Data were entered into EPI DATA 4.6 and exported to SPSS version 25 for further cleaning and analysis. Tables, figures, mean, and proportions are used to state the descriptive statistics of the study participants. The chi-square assumption was tested. Binary logistic regression was

fitted to identify independent predictors. Variables with a p-value of less than 0.25 in the bivariable logistic regression were included in the multivariable logistic regression to identify the final independent predictors. Hosmer-Lemeshow was used to test the model's goodness of fit. In addition, variance inflation factor (VIF), and standard error were used to screen multi-collinearity among independent variables. In the multivariable logistic regression analysis, a p-value of $\leq 0.05$ and 95% CI for the adjusted odds ratio (AOR) were used to determine the significance and degree of association between willingness to work in rural areas and the explanatory variables.

### Ethical considerations

The ethical clearance letter was obtained from the School of Midwifery under the delegation of the Institutional Review Board (IRB) of the University of Gondar. A formal administrative approval letter was gained from the college of medicine and health sciences. After clearly explaining the purpose of the study to the students, anonymous written informed consent was obtained from each study participant. Interviewees were given the right to refuse and withdraw from the study at any time.

## Result

### Socio-demographic characteristics of study participants

A total of 422 study participants were recruited but only 402 of them completed the study, giving a response rate of 95.26%. The mean age (standard deviation) of the students was 23.9 years old (± 2.28) and the majority (70.4%) of students were in the age group of < 25 years. Nearly two-thirds (62.4%) of the study participants were males. About 29.4% of the participants were medical students and 46% of them were from urban areas by birthplace (**Table 1**).

### Academic, medical, and behavior-related characteristics

Of the total study participants, nearly three-fourths (73.9%) believed that more job opportunities are available in urban areas. About 58.5% of the study participants joined their first departmental choice and got information regarding the department before campus entry. More than two-thirds (68.4%) and fourth-fifths (81.1%) of the study participants had a favorable attitude towards working in rural areas and intended to stay in their profession, respectively (**Table 2**).

### Willingness to work in rural areas and associated factors among graduating students

The willingness of students to work in rural areas was 78.4% (95% CI: 74.3, 82.4). In the bivariable logistic regression, sex, age, mother's educational status, father's educational status, place of birth, intention to continue with the profession, and attitude towards working in the rural areas were associated with willingness to work in rural areas. However, in the multivariable logistic regression analysis being male, intention to continue with the profession, having a mother with no formal education and attended primary education, and having a favorable attitude towards working in rural areas were variables significantly associated with willingness to work in rural areas.

Hence, the odds of having a good willingness to work in rural areas were two times higher among male students (AOR = 2.15; 95% CI: 1.17, 3.94) as compared to female students. Similarly, students who had an intention to continue with their profession were 2.5 times higher (AOR = 2.5; 95% CI: 1.28, 4.86) compared with those students who had no intention. Likewise,

**Table 1. Socio-demographic characteristics of study participants in the University of Gondar, college of medicine and health sciences, northwest Ethiopia, 2021 (n = 402).**

| Characteristics | Frequency | Percentage (%) |
|---|---|---|
| **Age in year** | | |
| <25 | 283 | 70.4 |
| ≥25 | 119 | 29.4 |
| **Sex** | 251 | 62.4 |
| Male | 151 | 37.6 |
| Female | | |
| **Place of birth** | | |
| Urban | 185 | 46 |
| Semi-urban | 59 | 14.7 |
| Rural | 158 | 39.3 |
| **Religion** | | |
| Orthodox Christian | 344 | 85.6 |
| Protestant | 42 | 10.4 |
| Others[a] | 16 | 4 |
| **Current marital status** | | |
| Married | 19 | 4.7 |
| In relationship | 31 | 7.7 |
| Single | 352 | 87.6 |
| **Average monthly pocket money** | | |
| < 500 ETB | 70 | 17.4 |
| 500–999 ETB | 108 | 26.7 |
| ≥ 1000 ETB | 224 | 55.7 |
| **Department** | | |
| Medicine | 118 | 29.4 |
| Nursing | 43 | 10.7 |
| Midwifery | 33 | 8.2 |
| Public health officer | 32 | 8 |
| Psychiatry | 17 | 4.2 |
| Environmental and occupational health | 21 | 5.2 |
| Optometry | 17 | 4.2 |
| Physiotherapy | 17 | 4.2 |
| Health informatics | 20 | 5 |
| Anesthesia | 18 | 4.5 |
| Pharmacy | 35 | 8.7 |
| Medical laboratory | 31 | 7.7 |
| **Average monthly income of the family** | | |
| <1500 ETB | 14 | 3.5 |
| 1501–5000 ETB | 115 | 28.6 |
| ≥ 5001 ETB | 273 | 67.9 |
| **Mothers educational status** | | |
| No formal education | 159 | 39.6 |
| Primary education | 83 | 20.6 |
| Secondary education | 67 | 16.7 |
| Diploma and above | 93 | 23.1 |
| **Mother's occupation** | | |
| Housewife and/or farmer | 229 | 57 |

*(Continued)*

**Table 1.** (Continued)

| Characteristics | Frequency | Percentage (%) |
|---|---|---|
| Government employee | 67 | 16.7 |
| Self employed | 47 | 11.7 |
| Merchant | 57 | 14.2 |
| Others[b] | 2 | 0.5 |
| **Father's educational status** | | |
| No formal education | 110 | 27.4 |
| Primary | 120 | 29.9 |
| Secondary | 47 | 11.7 |
| Diploma and above | 125 | 31.1 |
| **Fathers occupation** | | |
| Farmer | 168 | 41.8 |
| Merchant | 74 | 18.4 |
| Government employed | 105 | 26.1 |
| Self employed | 53 | 13.2 |
| Others[b] | 2 | 0.5 |

a = Muslim and catholic

b = retired, ETB- Ethiopian Birr

the likelihood of having a willingness to work in rural areas was 7.32 times higher (AOR = 7.32; 95% CI: 5.71, 18.65) among students who had a favorable attitude towards working in rural areas as compared to their counterparts. Lastly, students who had a mother with no formal education and attended primary education were 2.23 (AOR = 2.23; 95% CI: 1.02, 4.85) and 2.69 times (AOR = 2.69; 95% CI: 1.1, 6.61) more likely to have had a willingness to work in rural areas as compared to students who had a mother attended college and above education, respectively (**Table 3**).

## Perceived reasons for not willing to work in rural areas

About 21.6% (87) of the study participants were not willing to work in rural areas after graduation. The frequently mentioned reasons by the students for not to work in rural areas include lack of infrastructure in rural areas, not being far from family members, and peace and security issues (**Fig 2**).

## Discussion

This study aimed to assess the willingness to work in rural areas, and associated factors among graduating health students at the University of Gondar, northwest Ethiopia.

This study found that student's willingness to work in rural areas was 78.4%, which was far higher than studies conducted in Egypt-52.3% [14], China-52% [16], India-45% [17], Nigeria-33.5% [18], Ghana-55.4% [19], and Ethiopia-29.7% [20]. The possible explanation for the variation might be differences in socio-demographic contexts and time gaps. More than 80% of Ethiopia's population lives in rural areas [5], which may need more health workers, and new job opportunities in the health care system are available there. Besides, the number of health workers might be limited so far, and jobs might be available in the urban areas. Nowadays, however, there might be a higher number of health professionals than ever before, and jobs may not be easy to find, especially in Ethiopia. So newly graduating students may want to work in rural areas unreservedly. In light of this, the Ethiopian government and other

**Table 2. Academic, medical, and behavioral related characteristics of study participants in the University of Gondar, college of medicine and health sciences, northwest Ethiopia, 2021 (n = 402).**

| Characteristics | Frequency | Percentage (%) |
|---|---|---|
| **History of khat chewing** | | |
| Yes | 33 | 91.8 |
| No | 369 | 8.2 |
| **History of cigarette smoking** | | |
| Yes | 23 | 5.7 |
| No | 379 | 94.3 |
| **Ever drunk alcohol** | | |
| Yes | 178 | 55.7 |
| No | 224 | 44.3 |
| **Getting information about the department before campus entry** | | |
| Yes | 240 | 59.7 |
| No | 162 | 40.3 |
| **Source of information (n = 240)** | | |
| Friends | 116 | 48.3 |
| Healthcare providers | 17 | 7.1 |
| Families | 61 | 25.4 |
| Mass media | 46 | 19.2 |
| **Is the department your first choice** | | |
| Yes | 235 | 58.5 |
| No | 167 | 41.5 |
| **Intention to continue with the profession** | | |
| Yes | 329 | 81.1 |
| No | 73 | 18.2 |
| **Attitude towards working in rural areas** | | |
| Favorable | 275 | 68.4 |
| Unfavorable | 126 | 31.1 |
| **Where did you think jobs are available** | | |
| Urban | 297 | 73.9 |
| Rural | 105 | 26.1 |
| **Do you have any known medical disorder** | | |
| Yes | 384 | 95.5 |
| No | 18 | 4.5 |

stakeholders should address and implement conditions for job creation for health students, and arrange opportunities that will favor students to work in rural areas in all aspects. In turn, this idea will reduce the student's intention to leave their profession and burnout in the workplace.

Regarding factors associated with willingness to work in rural areas, this study affirmed that the sex of students affects their willingness to work in rural areas. Accordingly, male students were two times more likely to have had a good willingness to work in rural areas as compared to their female counterparts. This finding is consistent with studies conducted in Egypt and Ghana [16, 19]. This can be explained by the fact that male students might cope with problems that are considered to be present in rural areas, such as transportation problems, lack of electricity, and lack of safe water. When there is a lack of transportation, health workers may walk to the city on foot. Thus, females may be afraid that they will be in trouble. However, this finding contradicts a study conducted in China that founds female students intend to work more

**Table 3. Bivariable and multivariable logistic regression analysis of factors associated with graduating students willingness to work in rural areas in the University of Gondar, northwest Ethiopia, 2021 (n = 402).**

| Variables | Willingness to work in rural areas | | COR (95% CI) | AOR (95% CI) |
|---|---|---|---|---|
| | Yes | No | | |
| **Sex** | | | | |
| Male | 214 | 37 | 2.86 (1.76, 4.65) | **2.15 (1.17, 3.94)**[*] |
| Female | 101 | 50 | 1 | 1 |
| **Age in year** | | | | |
| < 25 | 213 | 70 | 1 | 1 |
| ≥ 25 | 102 | 17 | 1.97 (1.10, 3.52) | 1.35 (0.66, 2.75) |
| **Place of birth** | | | | |
| Urban | 132 | 53 | 1 | 1 |
| Semi-urban | 46 | 13 | 1.42 (0.71, 2.84) | 0.92 (0.36, 2.31) |
| Rural | 137 | 21 | 2.62 (1.49, 4.58) | 1.61 (0.63, 4.12) |
| **Intention to continue with the profession** | | | | |
| Yes | 267 | 62 | 2.24 (1.28, 3.91) | **2.50 (1.28, 4.86)**[*] |
| No | 48 | 25 | 1 | 1 |
| **Mother's educational status** | | | | |
| No formal education | 137 | 22 | 2.54 (1.35,4.81) | **2.23 (1.02, 4.85)**[*] |
| Primary education | 72 | 11 | 2.68 (1.23, 5.82) | **2.69 (1.1, 6.61)**[*] |
| Secondary education | 40 | 27 | 0.6 (0.31, 1.17) | 0.58 (0.268, 1.27) |
| Diploma and above | 66 | 27 | 1 | 1 |
| **Father's educational status** | | | | |
| No formal education | 96 | 14 | 3.35 (1.70, 6.56) | 3.35 (1.00, 10.36) |
| Primary education | 102 | 18 | 2.76 (1.48, 5.16) | 1.62 (0.66, 4.01) |
| Secondary education | 33 | 14 | 1.15 (0.55, 2.38) | 1.03 (0.39, 2.72) |
| Diploma and above | 84 | 41 | 1 | 1 |
| **Attitude towards working in the rural area** | | | | |
| Favorable | 249 | 26 | 8.7 (5.10, 14.85) | **7.32 (5.71,18.65)**[**] |
| Unfavorable | 66 | 60 | | 1 |

Notes

[*] P ≤ 0.05

[**]P ≤ 0.001.

Abbreviations: AOR, adjusted odds ratio; COR, crude odds ratio; CI, confidence interval; 1, reference category

in rural areas compared with their male counterparts [21]. This could be justified by differences in the country's infrastructure and culture of the community. In Ethiopia, the infrastructure might not be favorable for females like that of China. Besides, female students might not be willing to be far from families because of the perception of the community, which the community may not believe females can deal with the difficult environment.

This study revealed that students who had a mother with no formal education and had attended primary education were 2.23 and 2.69 times more likely to have had a willingness to work in rural areas as compared to students who had a mother who had attended college and above education, respectively. This finding is in agreement with a study conducted in China [21]. This could be explained by parents who might have a big influence on shaping and directing the attitude and behaviors of their children, including choosing departments and workplaces. Besides, those students who had a family with less educational achievement are expected to be less affluent. Thus, to support the household economy, students will have a willingness to work in rural areas irrespective of preconditions.

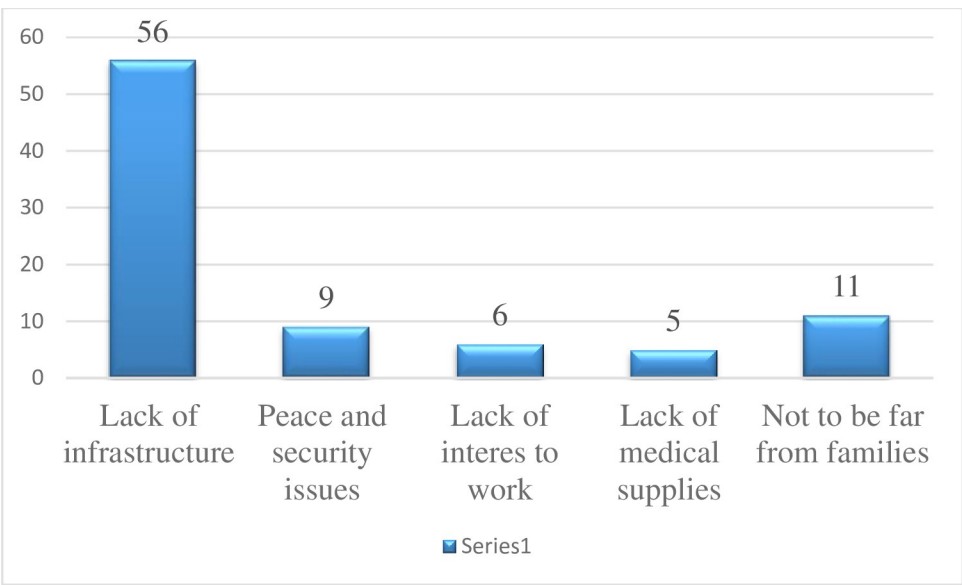

**Fig 2. Reasons why students are unwilling to work rural in rural areas at the University of Gondar, northwest Ethiopia, 2021.**

In this study, the odds of willingness to work in rural areas were two times higher among students who had plans to continue with their profession as compared to those students who had no intention to continue with the profession that they have currently. A study conducted in Nigeria reported a similar result [23]. This could be explained by the fact that students who intend to continue with their profession may initially join the department of their choice and could have a strong commitment to work in any part of the country. On the other hand, students may not have the interest to continue with their current profession due to different reasons like a plan to immigrate to foreign counties and engage in other business. Because those students with no intention to continue with their profession might have a plan to participate in other income-generating activities and plan to change their profession due to a lack of interest in the department after they are engaged. In this regard, the Ethiopian government should pay attention to the healthcare providers by settling different encouraging undertakings such as crafting job opportunities, evenhanded salary for their work, and timely educational opportunities for their vertical growth.

This study also found that students' attitude toward rural work affected their willingness to work in rural areas after graduation. Accordingly, students with a positive attitude towards working in rural areas were seven times more likely than students with unfavorable attitudes. This result is consistent with a study conducted in Ethiopia [15]. The attitude of an individual has a direct impact on determining responses to situations, psychological preparations, and influencing responses to all situations around them [17]. Thus, students who have a positive attitude to the need for rural community support and access to equitable health care may want to work in rural areas unconditionally. They might also be happy to help people, and able to fulfill their good wishes by serving the rural people, where highly in need of healthcare access. Lastly, we are pleased to acknowledge some of the limitations of this study. Due to the cross-sectional nature of the study design, it may not be possible to know the exact effect of the identified factors on the outcome variable. Besides, the study lumped all health professional together against to the WHO definition of healthcare providers for the rural community. However, medical, nursing, and midwifery professionals are highly needed by the rural

community. Moreover, since the questionnaire was self-administered, response bias might be introduced. However, before data collection, students were told that their participation in the research is significant and that they would fill in the required information honestly for the success of the research.

## Conclusion

More than three-quarters of health students were willing to work in rural areas. Being a male student, having the intention to stay in their profession, having a good attitude toward working in rural areas, and students having a mother of low academic achievement were observed to affect the student's willingness to work in the rural areas. Thus, this calls for health policy-makers and other stakeholders to give attention to arranging all-encompassing job opportunities in the rural areas that will favor both males and females. In addition, it is important to build a good view and understanding for students about rural areas during their campus life.

## Supporting information

**S1 Questionnaire. English and Amharic version.**
(DOCX)

**S1 Dataset. SPSS data.**
(SAV)

## Acknowledgments

We would like to thank the University of Gondar for providing study ethical clearance to conduct this study. Our gratitude also goes to all data collectors and study participants. We are glad to the University of Gondar for writing permission letter.

## Author Contributions

**Conceptualization:** Azmeraw Ambachew Kebede.

**Data curation:** Blen Getachew, Biruk Bizuneh, Birhanu Ewunetu, Dawit Kassahun, Dessalegn Fentahun, Destaw Ademe, Endeshaw Admasu Cherkos, Azmeraw Ambachew Kebede.

**Formal analysis:** Blen Getachew, Biruk Bizuneh, Birhanu Ewunetu, Dawit Kassahun, Dessalegn Fentahun, Destaw Ademe, Endeshaw Admasu Cherkos, Azmeraw Ambachew Kebede.

**Investigation:** Blen Getachew, Biruk Bizuneh, Birhanu Ewunetu, Dawit Kassahun, Dessalegn Fentahun, Destaw Ademe, Endeshaw Admasu Cherkos, Azmeraw Ambachew Kebede.

**Methodology:** Blen Getachew, Biruk Bizuneh, Birhanu Ewunetu, Dawit Kassahun, Dessalegn Fentahun, Destaw Ademe, Endeshaw Admasu Cherkos, Azmeraw Ambachew Kebede.

**Visualization:** Blen Getachew, Biruk Bizuneh, Birhanu Ewunetu, Dawit Kassahun, Dessalegn Fentahun, Destaw Ademe, Endeshaw Admasu Cherkos, Azmeraw Ambachew Kebede.

**Writing – original draft:** Blen Getachew, Azmeraw Ambachew Kebede.

**Writing – review & editing:** Blen Getachew, Biruk Bizuneh, Birhanu Ewunetu, Dawit Kassahun, Dessalegn Fentahun, Destaw Ademe, Endeshaw Admasu Cherkos, Azmeraw Ambachew Kebede.

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
