## [Decision Letter · Decision Letter 0]

5 Jul 2022

PONE-D-21-30311Willingness to work in rural areas and associated factors among graduating medical and health science students at the University of Gondar, northwest Ethiopia, 2021PLOS ONE

Dear Dr. Kebede,

Thank you for submitting your manuscript to PLOS ONE. After careful consideration, we feel that it has merit but does not fully meet PLOS ONE’s publication criteria as it currently stands. Therefore, we invite you to submit a revised version of the manuscript that addresses the points raised during the review process.

We have obtained reports from two reviewers who had important reservations with regards to some aspects of this manuscript. Their final recommendations were divergent and I recommend that the authors revise and resubmit, addressing all of the reviewers' comments individually.

We look forward to receiving your revised manuscript.

Kind regards,

Filipe Prazeres, MD, MSc, Ph.D.

Academic Editor

PLOS ONE

Journal Requirements:

2. Please include additional information regarding the survey or questionnaire used in the study and ensure that you have provided sufficient details that others could replicate the analyses. For instance, if you developed a questionnaire as part of this study and it is not under a copyright more restrictive than CC-BY, please include a copy, in both the original language and English, as Supporting Information. If the original language is written in non-Latin characters, for example Amharic, Chinese, or Korean, please use a file format that ensures these characters are visible.

4. Please upload a copy of Figure 2, to which you refer in your text on page 16. If the figure is no longer to be included as part of the submission please remove all reference to it within the text.

Reviewers' comments:

Reviewer's Responses to Questions

**Comments to the Author**

1. Is the manuscript technically sound, and do the data support the conclusions?

Reviewer #1: No

Reviewer #2: Partly

2. Has the statistical analysis been performed appropriately and rigorously? 

Reviewer #1: Yes

Reviewer #2: Yes

3. Have the authors made all data underlying the findings in their manuscript fully available?

Reviewer #1: Yes

Reviewer #2: Yes

4. Is the manuscript presented in an intelligible fashion and written in standard English?

Reviewer #1: No

Reviewer #2: No

5. Review Comments to the Author

Reviewer #1: Thank you for undertaking this important piece of research work in providing more insights into the issues affecting the distribution of health workers to rural areas.

My comments have been classified into major and minor issues and are as follows:

Major issues:

#1. I believe the use of the heterogeneous group of different cadres of health personnel is not adequate for the following reasons:

1.1 WHO in its world health report 2006 (World Health Organization. (‎2006)‎. The world health report : 2006 : working together for health) has clearly defined health professionals to include doctors, nurses and midwife in training hence comparison cannot be made lumping up other classes of health workers in training who ordinarily may not be heavily impacted by the issue under consideration given that their services are not always primarily needed as core areas for health provision mandate in rural areas.

1.2 The previous works cited and compared with your results clearly have defined their works with a specific group of class of health worker student’s usually medical students.

Given the difference in training, scope, primary job deployment (e.g. PHC workers for rural areas) and their need in rural areas amongst others, it may be difficult to lump all the classes of health workers student together.

Thus given the foregoing it may be necessary to substantially redo the recruitment and focus either on health professionals student - medical and nursing students or the other cadres who ordinarily are expected to be found in the rural areas in the PHC. Then the analysis will focus on either of the group with a new calculated sample size.

#2. Can you be more explicit and outline the attitude related questions making reference to the questionnaire in the method section? Again, a write up including some diagram or table for illustration demonstrating the derivation of the attitude summation scores from the question and its final categorization would be in order as already provided from the questionnaire.

#3. Why the use of 0.2 for test of significance for the bivariable, any reference or rationale for the choice?

What is the difference between bivariable logistic regression and chi-square and the multivariable logistic regression? How was the final model for the multivariable created ie which variables were selected and added into that final model? Did all the variables in the bivariabe meet the p-value cutoff you stated you used in moving to the multivariable regression?

What was only displayed in the table was the COR and AOR with pvalue only for the AOR. I think it may be good to make this much clearer.

#4 It would have been nice to add some write up on the results from the bivariable analysis before detailing on the multivariable adjusted ones

#5.There is a need to rework and format your reference list for consistency as some had missing authors name(eg ref. 23), more authors than the original paper(eg ref3), wrong citation(ref 2), typos, etc. It would also be nice to add the doi for those articles that have.

Minor Issues

#6 Please can you explain the choice of the use of 50% for previous prevalence (p) for the sample size calculation? Can you provide the reference or provide justification for the choice.

#7. There are lots of grammatical errors that may need the article to be copy edited to improve on its language structure. I have highlighted a few in the body of the article text.

#8.Is it stratified simple random sampling or stratified random sampling?

#9. Other minor issues and comments especially for grammatical issues are as contained in my comments and highlight in the article.

Reviewer #2: Very interesting article. This is a relevant social and medical issue that must be discussed.

I have some observations:

1- The keywords do not reflect the purpose of the article - for example, the MeSH term "Rural Health Services" could be included.

2- It's necessary to revise the text. For example:

- "The willingness of students to work in the rural areas was the outcome variable, while age, sex..." - Incomplete sentence

- "Students were asked whether they have a willingness to work in rural areas or not after graduation. Students who said “yes” when they are asked, whether they are willing to work in the rural areas after graduation or not, was considered..." - Repetition

- "who hadn’t have intention..." - Wrong past tense

- "Besides, the multivariable logistic regression analysis revealed that being male, had an intention to continue with the profession, had a mother with no formal education..." - Wrong tense

- "were variables having a significant association" - Wrong tense

- "This could be explained by parents might have a big influence on shaping and directing the attitude and behaviors of their children, including educations what to learn and work something else." - Educations what to learn?

- "as compared to those students who hadn’t intended to continue with the profession that they have currently..." - Wrong past tense

- "Because those students with no intention to continue with their profession might have a plan to participate..." - incomplete sentence

- "This study also was found that students’ attitudes" - Also was found?

- "having a mother of low academic achievement was observed to affect"  were observed

3- The first paragraph of the discussion was the repetition of the results.

4- In the conclusion, "having a good attitude towards working in rural areas" was repeated. Also "Moreover, benefits improvement is essential for students to stay in their profession without getting disappointed and planned to leave their profession." was not raised in the discussion.

5- Table 1: > 25 ≥25  I think it was < 25 and ≥25

The monthly income (Average monthly pocket money) is in what currency? Dollar?

6- Table 2: "chat chewing"  isn't khat chewing?

You don't need to put rows for "yes" and "no" in the table. You can just put "yes".

Example:

History of chat chewing, Yes 33 91.8

English must be revised. However, I reinforce the value of the article and thematic.

6. PLOS authors have the option to publish the peer review history of their article (what does this mean?). If published, this will include your full peer review and any attached files.

Reviewer #1: No

Reviewer #2: **Yes: **Lis Campos Ferreira

---

## [Decision Letter · Decision Letter 1]

15 Aug 2022

PONE-D-21-30311R1Willingness to work in rural areas and associated factors among graduating medical and health science students at the University of Gondar, northwest Ethiopia, 2021PLOS ONE

Dear Dr. Kebede,

Thank you for submitting your manuscript to PLOS ONE. After careful consideration, we feel that it has merit but does not fully meet PLOS ONE’s publication criteria as it currently stands. Therefore, we invite you to submit a revised version of the manuscript that addresses the points raised during the review process.

We look forward to receiving your revised manuscript.

Kind regards,

Filipe Prazeres, MD, MSc, Ph.D.

Academic Editor

PLOS ONE

Reviewers' comments:

Reviewer's Responses to Questions

**Comments to the Author**

1. If the authors have adequately addressed your comments raised in a previous round of review and you feel that this manuscript is now acceptable for publication, you may indicate that here to bypass the “Comments to the Author” section, enter your conflict of interest statement in the “Confidential to Editor” section, and submit your "Accept" recommendation.

Reviewer #1: (No Response)

Reviewer #2: All comments have been addressed

2. Is the manuscript technically sound, and do the data support the conclusions?

Reviewer #1: No

Reviewer #2: Yes

3. Has the statistical analysis been performed appropriately and rigorously? 

Reviewer #1: No

Reviewer #2: Yes

4. Have the authors made all data underlying the findings in their manuscript fully available?

Reviewer #1: Yes

Reviewer #2: Yes

5. Is the manuscript presented in an intelligible fashion and written in standard English?

Reviewer #1: Yes

Reviewer #2: Yes

6. Review Comments to the Author

Reviewer #1: Thank you for responding to some of the issues I earlier highlighted in the last review but I still have some outstanding concerns that may not have been adequately addressed.

My comments have been classified into major and minor issues and are as follows:

Major issues:

#1. In the earlier review I had drawn attention about my concern in lumping, together of the different diverse classes of students together which I find problematic based on the grounds I had raised earlier. I believe the lumping together makes it difficult in being able to pin point who exactly one is referring to as compared to previous works cited in the references and literature, which had a clear-cut respondents.

From the response to my earlier comments , I believe you have not provided any literature backing for the need to go in this way of classification as against the norm ie the need to lumping all such categories of students whose diversity and albinito expectations and categorization for rural health workforce diverge significantly.

Lumping all together as “medical and health students” appears too ambiguous, overarching, and unrepresentative of any specific group. I have struggled in finding any reference to the use of such terms in other studies in this area previously. Apparently, some of the categories of the students have different length of studies, different scopes and expectations thus making it too heterogeneous to be all lumped together. From earlier reports from a world bank report for Ethiopian health workers it had pointed out that albinitio , there are different expectations and willingness to working in rural areas for the different categories (Feysia, Berhanu & Herbst, Christopher & Lemma, Wuleta & Soucat, Agnes. (2012). The Health Workforce in Ethiopia: Addressing the Remaining Challenges. 10.13140/RG.2.2.25472.84482.).

This I believe is still a fundamental issue that has not yet been provided addressed. Is there any professional category that lumps together as “Medical and health students”?

I would strongly recommended that the work should pick specific categories of students that can be lumped together in line with the WHO health report 2006 classification or even based on the end classification of the profession based on high level or mid-level health manpower based on this world bank report ( Feysia, Berhanu & Herbst, Christopher & Lemma, Wuleta & Soucat, Agnes. (2012). The Health Workforce in Ethiopia: Addressing the Remaining Challenges. 10.13140/RG.2.2.25472.84482.)

This would enable the findings to represent a more homogenous group that the results can specifically related to. This would mean

I have brought forward my earlier comments here as an addendum to the above.

I believe the use of the heterogeneous group of different cadres of health personnel is not adequate for the following reasons:

1.1 WHO in its world health report 2006 (World Health Organization. (‎2006)‎. The world health report : 2006 : working together for health) has clearly defined health professionals to include doctors, nurses and midwife in training hence comparison cannot be made lumping up other classes of health workers in training who ordinarily may not be heavily impacted by the issue under consideration given that their services are not always primarily needed as core areas for health provision mandate in rural areas.

Than

1.2 The previous works cited and compared with your results clearly have defined their works with a specific group of class of health worker student’s usually medical students.

Given the difference in training, scope, primary job deployment (e.g. PHC workers for rural areas) and their need in rural areas amongst others, it may be difficult to lump all the classes of health workers student together.

Thus given the foregoing it may be necessary to substantially redo the recruitment and focus either on health professionals student - medical and nursing students or the other cadres who ordinarily are expected to be found in the rural areas in the PHC. Then the analysis will focus on either of the group with a new calculated sample size.

Minor Issues

#2 A minor edit on your reference 7 link as well as other references I have added as a comment in the Pdf.

Reviewer #2: Congratulations to the authors for the attention given to the reviewers' comments. The quality of the article has significantly improved.

7. PLOS authors have the option to publish the peer review history of their article (what does this mean?). If published, this will include your full peer review and any attached files.

Reviewer #1: No

Reviewer #2: **Yes: **Lis Campos Ferreira

---

## [Decision Letter · Decision Letter 2]

19 Sep 2022

PONE-D-21-30311R2Willingness to work in rural areas and associated factors among graduating medical and health science students at the University of Gondar, northwest Ethiopia, 2021PLOS ONE

Dear Dr. Kebede,

Thank you for submitting your manuscript to PLOS ONE. After careful consideration, we feel that it has merit but does not fully meet PLOS ONE’s publication criteria as it currently stands. Therefore, we invite you to submit a revised version of the manuscript that addresses the points raised during the review process.

We look forward to receiving your revised manuscript.

Kind regards,

Filipe Prazeres, MD, MSc, Ph.D.

Academic Editor

PLOS ONE

Journal Requirements:

Reviewers' comments:

Reviewer's Responses to Questions

**Comments to the Author**

1. If the authors have adequately addressed your comments raised in a previous round of review and you feel that this manuscript is now acceptable for publication, you may indicate that here to bypass the “Comments to the Author” section, enter your conflict of interest statement in the “Confidential to Editor” section, and submit your "Accept" recommendation.

Reviewer #1: (No Response)

2. Is the manuscript technically sound, and do the data support the conclusions?

Reviewer #1: Partly

3. Has the statistical analysis been performed appropriately and rigorously? 

Reviewer #1: Yes

4. Have the authors made all data underlying the findings in their manuscript fully available?

Reviewer #1: Yes

5. Is the manuscript presented in an intelligible fashion and written in standard English?

Reviewer #1: Yes

6. Review Comments to the Author

Reviewer #1: Thank you for responding to some of the issues I earlier highlighted in the last review but find below my few comments:

This are my comments:

1. I would differ slightly concerning the relevance of the WHO 2006 report which is still one of the fundamental documents when discussing around Human resource for Health issue. Even going through the report by Feysia, Berhanu & Herbst, Christopher & Lemma, Wuleta & Soucat, Agnes. (2012). The Health Workforce in Ethiopia: Addressing the Remaining Challenges. 10.13140/RG.2.2.25472.84482 and other seminal HRH papers produced by WHO and other organization some of the issue I have raised has remained.

Going forward I think it would be best you change the term `medical and health students` to simply `health students` including in the title and every other parts of the article.

For instance the title may now read “Willingness to work in rural areas and associated factors among graduating Health students at the University of Gondar, northwest Ethiopia, 2021”.

This is because fundamentally all the groups mentioned are health students.

This I believe will allow the reader not to be confused given the conjoined terms is not universally used. The reader can then understand that it encompasses all students studying in one or the other of the health sciences, which I believe is what your study was about.

2. In the limitation paragraph you mentioned “Besides, the study lumped all health professional together against to the WHO definition of healthcare providers for the rural community. However, all healthcare professionals…”

I am thinking would it not be better specifying students studying one of the health courses since your study participants are currently graduating students….

7. PLOS authors have the option to publish the peer review history of their article (what does this mean?). If published, this will include your full peer review and any attached files.

Reviewer #1: No

---

## [Decision Letter · Decision Letter 3]

11 Oct 2022

Willingness to work in rural areas and associated factors among graduating health students at the University of Gondar, northwest Ethiopia, 2021

PONE-D-21-30311R3

Dear Dr. Kebede,

We’re pleased to inform you that your manuscript has been judged scientifically suitable for publication and will be formally accepted for publication once it meets all outstanding technical requirements.

Kind regards,

Filipe Prazeres, MD, MSc, Ph.D.

Academic Editor

PLOS ONE

Additional Editor Comments (optional):

Reviewers' comments:

Reviewer's Responses to Questions

**Comments to the Author**

1. If the authors have adequately addressed your comments raised in a previous round of review and you feel that this manuscript is now acceptable for publication, you may indicate that here to bypass the “Comments to the Author” section, enter your conflict of interest statement in the “Confidential to Editor” section, and submit your "Accept" recommendation.

Reviewer #1: All comments have been addressed

2. Is the manuscript technically sound, and do the data support the conclusions?

Reviewer #1: Yes

3. Has the statistical analysis been performed appropriately and rigorously? 

Reviewer #1: Yes

4. Have the authors made all data underlying the findings in their manuscript fully available?

Reviewer #1: Yes

5. Is the manuscript presented in an intelligible fashion and written in standard English?

Reviewer #1: Yes

6. Review Comments to the Author

Reviewer #1: (No Response)

7. PLOS authors have the option to publish the peer review history of their article (what does this mean?). If published, this will include your full peer review and any attached files.

Reviewer #1: No

---

## [Editor Report · Acceptance letter]

14 Oct 2022

PONE-D-21-30311R3 

Willingness to work in rural areas and associated factors among graduating health students at the University of Gondar, northwest Ethiopia, 2021 

Dear Dr. Kebede:

I'm pleased to inform you that your manuscript has been deemed suitable for publication in PLOS ONE. Congratulations! Your manuscript is now with our production department. 

Kind regards, 

on behalf of

Prof. Filipe Prazeres 

Academic Editor

PLOS ONE